# Prevalence and Subtype Distribution of *Blastocystis* Isolated from School-Aged Children in the Thai-Myanmar Border, Ratchaburi Province, Thailand

**DOI:** 10.3390/ijerph20010204

**Published:** 2022-12-23

**Authors:** Amanee Abu, Chantira Sutthikornchai, Aongart Mahittikorn, Khuanchai Koompapong, Rachatawan Chiabchalard, Dumrongkiet Arthan, Ngamphol Soonthornworasiri, Supaluk Popruk

**Affiliations:** 1Department of Protozoology, Faculty of Tropical Medicine, Mahidol University, 420/6 Ratchawithi Road, Bangkok 10400, Thailand; 2Department of Tropical Nutrition and Food Science, Faculty of Tropical Medicine, Mahidol University, 420/6 Ratchawithi Road, Bangkok 10400, Thailand; 3Department of Tropical Hygiene, Faculty of Tropical Medicine, Mahidol University, 420/6 Ratchawithi Road, Bangkok 10400, Thailand

**Keywords:** *Blastocystis*, subtype, border area, school-aged children

## Abstract

*Blastocystis* is one of the most common enteric protozoa that inhabits the intestinal tract of humans and different animals. Moreover, it has a worldwide geographic distribution. Its main mode of transmission is via the fecal-oral route. At present, 26 subtypes are widely distributed across both humans and animals. The current study aimed to determine the prevalence and subtype distribution of *Blastocystis* among school-aged children living on the Thai-Myanmar border, Ratchaburi province, Thailand. In total, 508 samples were collected from children at six schools. The prevalence of *Blastocystis* infection was amplified and sequenced in the 600 bp barcode region of the small-subunit ribosomal RNA (SSU rRNA). The overall prevalence of *Blastocystis* infection was 3.35% (17/508). ST3 (11/17) was the most predominant subtype, followed by ST1 (5/17) and ST2 (1/17). A phylogenetic tree was constructed based on the Tamura92+G+I model using the maximum-likelihood algorithm. Surprisingly, all sequences of the ST3-positive samples were closely correlated with the cattle-derived sequence. Meanwhile, all sequences of the *Blastocystis* ST1-positive samples were closely correlated with the human-derived sequence. Nevertheless, further studies should be conducted to validate the zoonotic transmission of *Blastocystis*. Based on our findings, personal hygiene and sanitation should be improved to promote better health in children in this area.

## 1. Introduction

Intestinal parasitic infections are one of the main public health issues worldwide, and their incidence is higher in developing than in developed countries [1,2]. *Blastocystis* is a common enteric protozoan causing parasitic intestinal infections among children. Further, it mainly colonizes the gastrointestinal tract, and it is transmitted via the fecal-oral route [3,4]. At present, approximately more than 1 billion humans are infected with *Blastocystis* worldwide [5]. Its transmission is correlated with several factors, such as poor personal hygiene, consumption of contaminated food, and environmental conditions associated with poor quality of life [6,7,8]. Moreover, close contact with animals due to cultural conditions may cause *Blastocystis* infection [9,10,11]. Additionally, school-aged children are at high risk of *Blastocystis* infection due to poor hygiene habits [1,4]. Several studies showed that the prevalence of *Blastocystis* infection varies from 0.7% to 45.2% among school-aged children in Thailand [12,13,14,15]. At present, 26 subtypes of *Blastocystis* are widely distributed in both humans and animals. Among them, subtype 3 is most commonly observed in humans [16,17]. The methods used to detect *Blastocystis* are usually based on microscopy. However, these techniques have low sensitivity and specificity, and assessment using these strategies should be performed by skilled microscopists to validate morphology [18,19,20]. In recent years, polymerase chain reaction (PCR), a molecular biology technique, targets a region of the SSU rRNA genes. Moreover, it has the highest sensitivity and specificity and can identify different subtypes [21,22,23]. However, studies on the incidence and subtypes of *Blastocystis* in school-aged children living in the border areas of Thailand are limited. The current study aimed to determine the prevalence and subtype distribution of *Blastocystis* isolated from school-aged children living in the Thai-Myanmar border areas, Ratchaburi Province, Thailand.

## 2. Materials and Methods

### 2.1. Study Area

This cross-sectional study included children aged 4–12 years from six primary schools in Suan Pheung district, Ratchaburi province, near the Thai-Myanmar border, a natural border in western Thailand (Figure 1). Ratchaburi province is divided into 10 districts. Suan Pheung district, which has important natural resources, is the most popular. The forest covers an area of 1711 km^2^, or 33% of the provincial area. The area is part of the Tenasserim Hills and the Tanintharyi division of Myanmar. The population in this area have low income and education levels. Most people are agriculture employees and construction laborers and some are unemployed. They commonly consume rainwater, stream water, and water from community wells. Fecal samples were collected from children from HP School, TKL School, RJ School, TMK School, SP School, and BW School, which are located in different sub-districts and are all government primary schools.

### 2.2. Study Population and Sample Collection

The study protocol was approved by the Ethics Committee of the Faculty of Tropical Medicine, Mahidol University (MUTM: 2022-011-01). All fecal samples were collected from both boys and girls. The participants were commonly from the Kayin, Kayar, and Mon ethnic groups. They were aged between 4–12 years and were in kindergarten and grades 1–3. All participants were instructed to ask permission from their parents, and a written informed consent form was obtained before study participation. Children who were not allowed by their parents to join the research were excluded. In total, 508 fecal samples (*n* = 270, boys and *n* = 238, girls) were collected. Moreover, according to the schools, the numbers of samples collected were as follows: 98, TKL; 170, HP; 74, TMK; 96, SP; 47, BW; and 23, RJ. There was no substantial difference in the dates of fecal collection among schools. A single sample was collected from each participant. All participants were instructed to collect fecal samples properly. Clean and labeled plastic containers, toilet tissue paper, and applicator sticks were provided. During transportation, the fecal samples were kept preserved at −20 °C until DNA extraction.

### 2.3. Extraction of Genomic DNA from Fecal Samples

DNA was extracted from all fecal samples using a commercially available DNA extraction kit (PSP Spin Stool Kit, Stractec Molecular, Berlin, Germany), according to the manufacturer’s instructions. All extracted DNAs from fecal samples were stored at −20 °C until conventional PCR. The primers amplified ~600 bp of the SSU rRNA gene. The primers were as follows: forward primer RD5 5′-ATC TGG TTG ATC CTG CCA GT-3′ and reverse primer BhRDr 5′-GAG CTT TTTA ACT GCA ACA ACG-3′ [23]. The 25 µL reaction in each mixture tube contained 1x PCR buffer, 1.5 mM MgCl_2_, 0.2 mM dNTPs, 1 µM of each primer, and 2.5 U *Taq* DNA polymerase (Fermentus, Waltham, MA, USA). The PCR conditions were as follows: 1 cycle denaturing step at 94 °C for 3 min, followed by 35 cycles of annealing step at 59 °C for 1 min, and extension step at 72 °C for 1 min. The last cycle was extended for 5 min at 72 °C. The PCR product and 100bp DNA marker was run on 1.5% agarose gel electrophoresis using 1% tris-acetate-EDTA, which is a running buffer. The gel was stained with Hydra green and visualized under an ultraviolet transilluminator.

### 2.4. Sequencing and Phylogenic Analysis

*Blastocystis*-positive samples were sequenced on an ABI 3730x1 automated DNA sequencer. The nucleotide sequences were aligned and compared with reference sequences available in the GenBank database using BLAST (http://blast.ncbi.nlm.nih.gov/, accessed on 5 May 2022) for subtype identification. *The blastocystis* 18S allele database was tested at http://pubmlst.org/blastocystis/, accessed on 10 May 2022). All 17 nucleotide sequences and reference sequences (ST1-ST17) were manually edited using BioEdit version 7.2.5 (Ibis Biosciences, Inc., Carlsbad, CA, USA) and multiply aligned using the ClustalW programs. Phylogenetic analysis was performed using MEGA version 7. A phylogenetic tree was constructed using the Tamura92+G+I model with the maximum-likelihood algorithm and was tested with 1000 bootstrap replicates. 

### 2.5. Statistical Analysis

Descriptive analysis was performed to express the percentages of the positive samples of *Blastocystis*. The chi-square test was used to compare *Blastocystis* prevalence according to age and sex. A *p*-value of <0.05 was considered statistically significant.

## 3. Results

### 3.1. Prevalence of Blastocystis Infection

The prevalence rate of *Blastocystis* infection was 3.35% (17/508) based on the assessment performed using PCR, as shown in Table 1. According to the schools, the prevalence rate was 7.14% (7/98) in TKL, 5.40% (4/74) in TMK, 2.35% (4/170) in HP, 2.13% (1/47) in BW, 1.04% (1/96) in SP, and 0% (0/23%) in RJ.

### 3.2. Subtype Distribution of Blastocystis in School-Aged Children

The subtype of *Blastocystis* was determined via direct sequencing of the barcode region of the 18S rRNA gene. ST1, ST2, and ST3 were found in *Blastocystis*-positive fecal samples. ST3 was the predominant subtype (64.71%; 11/17), followed by ST1 (29.41%; 5/17) and ST2 (5.88%; 1/17). The *Blastocystis* prevalence was classified according to sex male and female; *χ*^2^ = 0.096, d*f* = 2, *p >* 0.05) and age ≤6 and >6 years; *χ*^2^ = 0.145, d*f* = 2, *p* > 0.05). Results showed that there were no significant differences in the *Blastocystis* prevalence between sex and age.

### 3.3. Accession Number of Positive Samples and Phylogenic Analysis

The 17 nucleotide sequences of the barcode region of the SSU rRNA gene were significantly similar (≥98%) to the references of *Blastocystis* deposited in the GenBank database Table 2. Results showed that the nucleotide sequence samples belong to the three district subtypes: ST3: TKL25, TKL58, TKL85, TKL95, TKL98, HP83, TMK10, TMK33, TMK69, and BW38; ST1: TKL67, TKL89, HP110, HP161, TMK55, and SP4; and ST2: HP95. The phylogenic tree of the 17 nucleotide sequence samples and the 17 reference sequences in each subtype in the GenBank database was constructed based on the Tamura92+G+I model using the maximum-likelihood algorithm. *Proteromonas lacerate* (U37108) and *Karotomorpha* spp. (DQ431242) were classified under the out-group. The sequence of *Blastocystis* ST1-positive samples was closely correlated with human-derived sequences in the GenBank database. Moreover, the sequence of the *Blastocystis* ST3-positive samples was closely correlated with that of the cattle-derived sequences in the GenBank database, as shown in Figure 2.

## 4. Discussion

*Blastocystis* is one of the most common intestinal protozoa that can infect both humans and animals, and it has a worldwide geographic distribution. The current study aimed to assess the prevalence and subtype distribution of *Blastocystis* in school-aged children living in the Thai-Myanmar border area. The present study showed a rather low prevalence of *Blastocystis* infection but higher than the study conducted on Thailand’s school children, such as in the Kanchanaburi provinces in the western region of Thailand [24]. It might be different from the method of detection and study population. In Thailand, the prevalence rates of *Blastocystis* infection are 0.7% to 45.2% in school-aged children and the highest infection in young children living in orphan homes in Pathum Thani province [12,13,14,15]. Several studies reported that the prevalence of *Blastocystis* infection was higher in developing countries than in developed countries. Moreover, the prevalence varies from 0.5% to 30% in developed countries, and it is as high as 100% in developing countries [25,26,27,28,29,30].

The border area in some countries is predominantly rural, people live in poverty, and they lack education. Moreover, its economic development is low [31,32], and the open-defecation behavior of people in this region is attributed to a lack of latrines and water; hence, they are at risk of infection [31,32]. The prevalence rates of *Blastocystis* are 37.2% on the Thai-Myanmar border and 6.29% on the China-Myanmar border. The infection is correlated with poor hygiene habits and sanitation in the communities, zoonotic transmission, person-to-person contamination, and consumption of contaminated water [31,32]. However, in this study, the prevalence of *Blastocystis* infection was low. This might be attributed to improvements in common living conditions and easy access to healthcare services, which are attributed to economic growth. Generally, children in this area are not Thai. Most of them are Kayin, Kayar, and Mon. They have access to education and have more opportunities for learning and healthy living. This might explain why the prevalence of *Blastocystis* infection was low.

Several methods are used for detecting *Blastocystis* in routine parasitology laboratory examinations, and these include microscopy, xenic in-vitro culture, and molecular biology-based techniques [18,19,20]. Detection is commonly based on microscopy. However, this method has low sensitivity and specificity. It may lead to underdiagnosis and false-positive or false-negative results. The assessment using this strategy should be performed by skilled microscopists to confirm the morphology [18,19,20]. Previous studies have shown that PCR is a rapid and effective method for identifying microorganisms. Further, it can be used for epidemiological studies with a large number of samples [33]. In this study, PCR and the direct sequence technique were applied to detect the subtype of *Blastocystis* using 600 bp of the barcode region. This is the best representative in public databases, and it can be utilized for the analysis of 18S alleles to identify intrasubtype genetic variations [23]. Currently, 26 subtypes of *Blastocystis* have been identified in humans and animals [16,34]. ST1-ST4 are the most common subtypes found in humans, and they can occur worldwide [35,36]. This study showed that ST3 was the predominant subtype in boys and girls, followed by ST1 and ST2. This result was similar to that of previous studies in Thailand [35,36] and others in different countries, such as Malaysia [37], Turkey [38], Peru [39], and Italy [40]. Our finding showed 18S alleles of *Blastocystis* in each ST, which include allele 4 (ST1), allele 9 (ST2), and alleles 34 and 37 (ST3). Moreover, allele 4 (ST1) is closely correlated with person-to-person transmission [41]. Our result followed that of a previous study on young children from daycare centers because children are less careful about their personal hygiene, leading to more person-to-person transmission [41]. Additionally, allele 9 was most prevalent in both humans and animals, and it might be correlated with zoonotic transmission [42]. Moreover, allele 34 was the predominant allele in non-human primates, such as cattle, thus indicating the possibility of non-host specificity [43]. The parents of most children were agriculture and animal-care workers. Hence, children could have close contact with animals that may cause zoonotic infection. Nevertheless, the main point of *Blastocystis* infection in children commonly depends on personal hygiene habits and sanitation. However, allele analysis can provide useful information about host specificity and geographic distribution, which can lead to a better understanding of *Blastocystis* transmission [42,43]. This study did not show a significant difference in the prevalence of *Blastocystis* infection according to age and sex, which was similar to the finding of the previous studies [44,45]. 

Based on the phylogenetic trees of the 17 nucleotide sequences collected from the samples, this protozoan belonged to the ST1, ST2, and ST3 subtypes. Surprisingly, our findings revealed that the nucleotide sequence samples of ST3 were closely correlated with the cattle-derived sequence of the reference sequence (GenBank database). Meanwhile, the nucleotide sequence samples of ST1 were significantly with human-derived sequences in the reference sequence (GenBank database). However, our findings of *Blastocystis* ST3 might be transmitted from animals to humans, which indicates a zoonotic transmission pattern in this area [23].

## 5. Conclusions

This study is the first regarding the prevalence and subtype distribution of *Blastocystis* sp. in school-aged children living in the Thai-Myanmar border area, Ratchaburi Province, western Thailand. The prevalence of *Blastocystis* infection was quite low in school-aged children. However, the zoonotic potential is considered in *Blastocystis* ST3 infection. Moreover, the transmission of *Blastocystis* infection between humans to humans should not be disregarded. In this study, no questionnaire was used to assess potential sources of infection. More robust approaches must be carried out to determine possible sources of infection. The risk factors that be studied include the use of unsafe water supplies in the locality, domestic and wild animals in the area, as well as transmission in the households of the school children. Moreover, personal hygiene habits should be improved to promote better health and quality of life among children. Further, the disposal of animal and human wastes should be managed properly.

## Figures and Tables

**Figure 1 ijerph-20-00204-f001:**
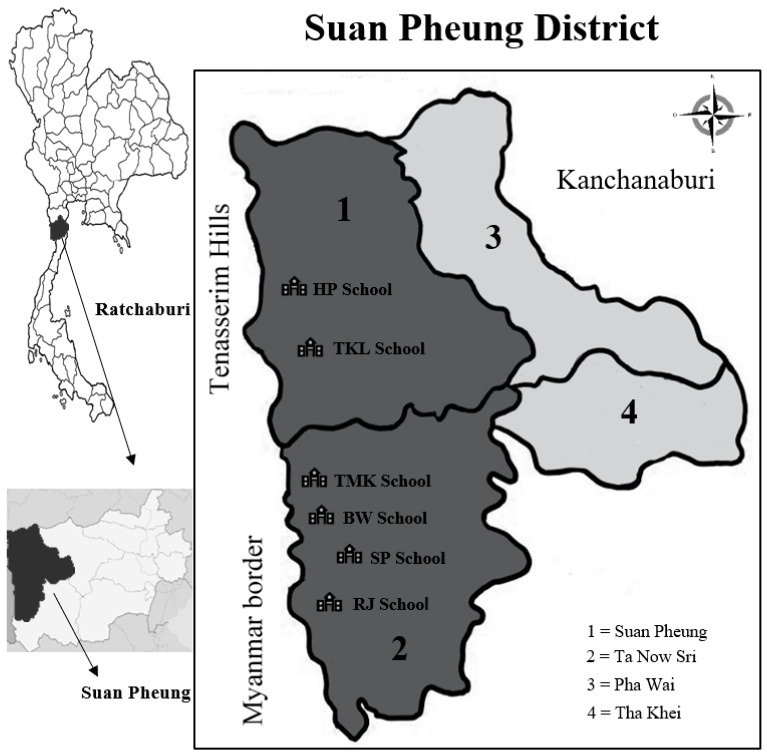
Map of the study area (modified from Wikipedia, 2010).

**Figure 2 ijerph-20-00204-f002:**
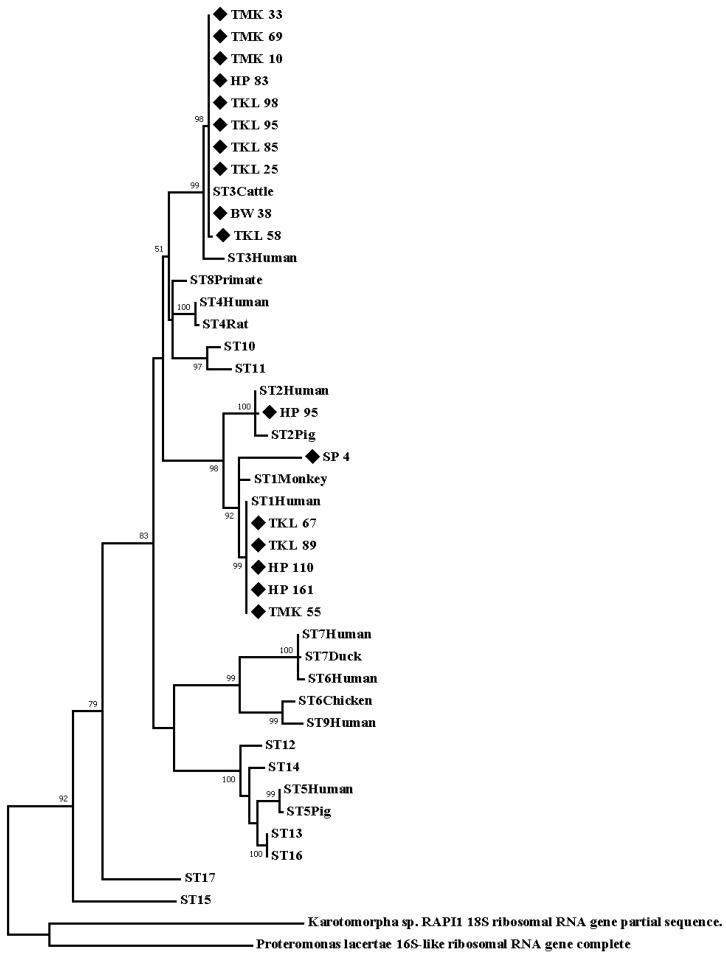
A phylogenic tree of the 17 nucleotide sequences was constructed based on theTamura92+G+I model using the maximum-likelihood algorithm.

**Table 1 ijerph-20-00204-t001:** Prevalence of *Blastocystis* infection in school-aged children.

School	Participants/Total No. (Kindergarten & Grades 1–3) (%)	Number of Positive Samples (%)
TKL School	98/239 (41.0)	7/98 (7.14)
HP School	170/228 (74.56)	4/170 (2.35)
TMK School	74/283 (26.15)	4/74 (5.40)
BW School	47/168 (27.98)	1/47 (2.13)
SP School	96/213 (45.07)	1/96 (1.04)
RJ School	23/85 (27.06)	0/23 (0)
Total	508/1216 (41.78)	17/508 (3.35%)

**Table 2 ijerph-20-00204-t002:** Accession numbers of positive samples used in the phylogenetic reconstruction.

Samples	Genbank Accession Number	Sex	Age	*Blastocystis* Subtype	Allele	Sequence Similarity (%)	Similar Genbank Reference Sequences (Source)
TKL25	OM149811	F	6	3	34	99.64%	MT039586 (Human)
TKL58	OM149812	M	7	3	36	99.64%	MT645665 (Human)
TKL67	OM149813	M	8	3	34	99.64%	MT645669 (Human)
TKL85	OM149814	F	8	3	34	99.44%	MT039597 (Human)
TKL89	OM149815	F	11	1	4	99.63%	MT645664 (Human)
TKL95	OM149816	F	8	3	34	98.92%	KY675330 (Human)
TKL98	OM149817	F	9	3	34	99.63%	MT039597 (Human)
HP83	OM149818	M	6	3	34	99.45	MG011642 (Human)
HP85	OM149819	F	6	2	9	99.11	MT039594 (Human)
HP110	OM149820	M	7	1	4	99.28	MT042818 (Human)
HP161	OM149822	F	6	1	4	100.00	MK719642 (Human)
TMK10	OM149823	M	5	3	34	99.28	MK100354 (Human)
TMK33	OM149824	M	7	3	34	99.82	MT039556 (Human)
TMK55	OM149825	M	9	1	4	99.64	MG011607 (Human)
TMK69	OM149826	M	5	3	34	100.00	MG011642 (Human)
BW38	OM149827	M	6	3	34	100.00	MT039567 (Human)
SP4	OM149828	M	9	1	4	99.80	MH021854 (Rat)

## Data Availability

Not applicable.

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
