# Peer review of "Prevalence and Subtype Distribution of Blastocystis Isolated from School-Aged Children in the Thai-Myanmar Border, Ratchaburi Province, Thailand"

_ijerph, 2022, doi:10.3390/ijerph20010204_

Round 1

Reviewer 1 Report

Study has scientific relevance and waswas well conducted. However, I have some questions to be clarified before the publication of this manuscript.

In discussion section:

* Improve the discussion on the application of diagnostic techniques with low sensitivity and specificity in the routine parasitology laboratory;

*Highlight for need for training professionals in clinical parasitology.

This fact is reflected in:

* false negative results

* false positive results

* underdiagnosis of  Blastocystis spp.

Author Response

Response to reviewers and editors

We would like to thank the reviewers and editors for their constructive comments and suggestions. We have considered all comments and suggestions during the revision of the manuscript and have made the suggested changes.

Below are our itemized responses (comments and suggestions of the reviewer first, followed by our responses).

Reviewer1

In the discussion section:

* Improve the discussion on the application of diagnostic techniques with low sensitivity and specificity in the routine parasitology laboratory;

*Highlight for the need for training professionals in clinical parasitology.

This fact is reflected in:

* false negative results

* false positive results

* underdiagnosis of  Blastocystis spp.

Answer: Thank you for your valuable comments and suggestions. We added the sentence “Detection is commonly based on microscopy. However, this method has low sensitivity and specificity. It may lead to underdiagnosis, false-positive or false-negative results. The assessment using this strategy should be performed by skilled microscopists to confirm the morphology”, page8.

Reviewer2

Line 21. May be more informative if the total population of the 6 schools surveyed is provided to give an indication of what percentage the 508 samples represent.

Answer: Thank you for your valuable comments and suggestions. We added information, as shown in Table 1.

Lines 67-69. When were the fecal samples collected from the different schools? Was there a substantial difference in the dates of collection between schools? Were single samples collected from each participating child?

Answer: Thank you for your valuable comments and suggestions. We added the sentence “There was no substantial difference in the dates of fecal collection among schools. A single sample was collected from each participant”, page3.

Lines 80-82. The total population of school children in each of the six participating schools should be given to determine the % of children participating in the study.

Answer: Thank you for your valuable comments and suggestions. We added information, as shown in Table 1.

 In Table 1 It would be more informative if the total school population from each of the participating schools as well as the % participating in the fecal examination were given. Another column can be added to incorporate this.

Answer: Thank you for your valuable comments and suggestions. We added information, as shown in Table 1.

Table 2. There is no need for Table 2 as the information can be included in the text.

Answer: Thank you for your valuable comments and suggestions. We deleted Table 2, according to your suggestion.

Table 3 is not needed as the information can be included in the text.

Answer: Thank you for your valuable comments and suggestions. We deleted Table 3, according to your suggestion.

Lines 217-219. The sentence must be revised for clarity. Although no questionnaire was used to assess potential sources of infection, more robust approaches must be carried out to determine possible sources of infection. The risk factors that be studied include the use of unsafe water supplies in the locality, domestic and wild animals in the area, as well as transmission in the households of the school children. These are some of the limitations of the study and should be stated as such in the conclusion.

Answer: Thank you for your valuable comments and suggestions. We add the sentence “In this study, no questionnaire was used to assess potential sources of infection, more robust approaches must be carried out to determine possible sources of infection. The risk factors that be studied include the use of unsafe water supplies in the locality, domestic and wild animals in the area, as well as transmission in the households of the school children.”, as shown in the conclusion.

Reviewer 2 Report

Line 21. May be more informative if the total population of the 6 schools surveyed is provided to give an indication of what percentage the 508 samples represent.

Lines 67-69. When were the fecal samples collected from the different schools? Was there a substantial difference in the dates of collection between schools? Were single samples collected from each participating child?

Lines 80-82. The total population of school children in each of the six participating schools should be given to determine the % of children participating in the study.

 In Table 1 It would be more informative if the total school population from each of the participating schools as well as the % participating in the fecal examination were given. Another column can be added to incorporate this.

Table 2. There is no need for Table 2 as the information can be included in the text.

Table 3 is not needed as the information can be included in the text.

Lines 217-219. The sentence must be revised for clarity. Although no questionnaire was used to assess potential sources of infection, more robust approaches must be carried out to determine possible sources of infection. The risk factors that be studied include the use of unsafe water supplies in the locality, domestic and wild animals in the area, as well as transmission in the households of the school children. These are some of the limitations of the study and should be stated as such in the conclusion.

Author Response

Response to reviewers and editors

We would like to thank the reviewers and editors for their constructive comments and suggestions. We have considered all comments and suggestions during the revision of the manuscript and have made the suggested changes.

Below are our itemized responses (comments and suggestions of the reviewer first, followed by our responses).

Reviewer1

In the discussion section:

* Improve the discussion on the application of diagnostic techniques with low sensitivity and specificity in the routine parasitology laboratory;

*Highlight for the need for training professionals in clinical parasitology.

This fact is reflected in:

* false negative results

* false positive results

* underdiagnosis of  Blastocystis spp.

Answer: Thank you for your valuable comments and suggestions. We added the sentence “Detection is commonly based on microscopy. However, this method has low sensitivity and specificity. It may lead to underdiagnosis, false-positive or false-negative results. The assessment using this strategy should be performed by skilled microscopists to confirm the morphology”, page8.

Reviewer2

Line 21. May be more informative if the total population of the 6 schools surveyed is provided to give an indication of what percentage the 508 samples represent.

Answer: Thank you for your valuable comments and suggestions. We added information, as shown in Table 1.

Lines 67-69. When were the fecal samples collected from the different schools? Was there a substantial difference in the dates of collection between schools? Were single samples collected from each participating child?

Answer: Thank you for your valuable comments and suggestions. We added the sentence “There was no substantial difference in the dates of fecal collection among schools. A single sample was collected from each participant”, page3.

Lines 80-82. The total population of school children in each of the six participating schools should be given to determine the % of children participating in the study.

Answer: Thank you for your valuable comments and suggestions. We added information, as shown in Table 1.

 In Table 1 It would be more informative if the total school population from each of the participating schools as well as the % participating in the fecal examination were given. Another column can be added to incorporate this.

Answer: Thank you for your valuable comments and suggestions. We added information, as shown in Table 1.

Table 2. There is no need for Table 2 as the information can be included in the text.

Answer: Thank you for your valuable comments and suggestions. We deleted Table 2, according to your suggestion.

Table 3 is not needed as the information can be included in the text.

Answer: Thank you for your valuable comments and suggestions. We deleted Table 3, according to your suggestion.

Lines 217-219. The sentence must be revised for clarity. Although no questionnaire was used to assess potential sources of infection, more robust approaches must be carried out to determine possible sources of infection. The risk factors that be studied include the use of unsafe water supplies in the locality, domestic and wild animals in the area, as well as transmission in the households of the school children. These are some of the limitations of the study and should be stated as such in the conclusion.

Answer: Thank you for your valuable comments and suggestions. We add the sentences “In this study, no questionnaire was used to assess potential sources of infection, more robust approaches must be carried out to determine possible sources of infection. The risk factors that be studied include the use of unsafe water supplies in the locality, domestic and wild animals in the area, as well as transmission in the households of the school children.”, as shown in the conclusion.